# Magnesium Fertilization Increases Nitrogen Use Efficiency in Winter Wheat (*Triticum aestivum* L.)

**DOI:** 10.3390/plants11192600

**Published:** 2022-10-02

**Authors:** Jarosław Potarzycki, Witold Grzebisz, Witold Szczepaniak

**Affiliations:** Department of Agricultural Chemistry and Environmental Biogeochemistry, Poznan University of Life Sciences, Wojska Polskiego 28, 60-637 Poznan, Poland

**Keywords:** in-soil Mg application, foliar Mg fertilization, Mg uptake, N uptake, nutrient use efficiency indices

## Abstract

Wheat fertilized with Mg, regardless of the method of application, increases nitrogen fertilizer (N_f_) efficiency. This hypothesis was tested in 2013, 2014, and 2015. A two-factorial experiment with three doses of Mg (i.e., 0, 25, and 50 kg ha^−1^) and two stages of Mg foliar fertilization (without; BBCH 30; 49/50; 30 + 49/50) was carried out. Foliar vs. in-soil Mg fertilization resulted in a comparable grain yield increase (0.5–0.6 t ha^−1^). The interaction of both fertilization systems increased the yield by 0.85–0.9 t ha^−1^. The booting/heading phase was optimal for foliar fertilization. Mg accumulation by wheat fertilized with Mg increased by 17% compared to the NPK plot. The recovery of foliar Mg was multiple in relation to its dose. The recovery of the in-soil Mg applied ranged from 10 to 40%. The increase in yield resulted from the effective use of N taken up by wheat. In 2014 and 2015, this amount was 21–25 kg N ha^−1^. The increase in yield resulted from the extended transfer of N from vegetative wheat parts to grain. Mg applied to wheat, irrespective of the method, increased the efficiency of the N taken up by the crop. Mg fertilization resulted in higher N_f_ productivity, as indicated by the increased nitrogen apparent efficiency indices.

## 1. Introduction

Wheat is a basic source of staple food for the world’s human population. Currently, the largest producers are China, India, the Russian Federation, the USA, Canada, France, and Ukraine. The mean yields in these countries in 2018–2021 were 5.6 ± 0.17, 3.4 ± 0.12, 2.8 ± 0.15, 3.9 ± 0.23, 3.4 ± 0.01, 7.1 ± 0.58, and 3.3 ± 0.14, respectively [1]. The yield variability, as shown by the coefficient of variation (CV), in leading producers such as Canada, China, and India was 3.5%, 3.0%, and 2.5%, respectively. This low CV indicates an extensive system of wheat production. Therefore, in these countries, the yield gap (YG), as a measure of the ineffectiveness of the applied means of production, is low [2]. For Germany, the average yield in 2018–2020 was 7.3 ± 0.58 t ha^−1^, compared to 4.5 ± 0.65 t ha^−1^ for Poland. The CV for Germany was 7.9% and for Poland 14.4%. 

A realistic determination of the maximum attainable yield, as the basis for calculating the YG, requires local data that reflect both the short-term variability in weather conditions and the variability in management factors affecting the actual yield. The order of these factors for wheat in Poland, in descending order, is nitrogen (N), fore-crop, soil class, available potassium content, crop protection level—fungicide and foliar fertilization, and available phosphorus content [3]. Another method for the calculation of the YG, which is, in fact, simpler to use, is the concept of the nitrogen gap [4,5]. Both methods allow for the discrimination of factors, which is critical for the maximum wheat yield in a well-defined geographical area [6]. 

The nutrient requirements of wheat compared to other cereals is high, including, firstly, N, P, and K [7]. The yield-forming function of N is widely discussed in both science and agricultural practice. Some of the most important data on the impact of N on yield in wheat concern the *critical window* that extends from the beginning of stem elongation to flowering. N supply to wheat during this period is critical for both the number of ears and the number of grains per ear. Both yield components determine the aggregate component of the yield, i.e., the number of grains per unit area (grain density (GD) [8,9]. 

New, high-yielding wheat varieties developed to exploit their yield potential, require, first, the development of efficient technologies aimed at the effective use of nitrogen fertilizer (N_f_). The price of N_f_ increased many times in 2021 and 2002 [10]. The 4R Stewardship approach, the keywords of which are right source, right rate, right time, and right place, is actually limited by N_f_ [11]. 

Magnesium (Mg) cannot be treated as a *forgotten* nutrient, especially in modern, intensive, and effective agriculture [12,13]. The physiological and yield-forming functions of Mg are well known [14,15,16]. The increase in the yield of cereals in response to the application of Mg ranges from 5 to 10% [17]. The Mg content in the flag leaf of winter wheat at the onset of plant flowering can be used to forecast the yield [18]. The amount of Mg accumulated in crop plants is relatively low compared even to phosphorus [14]. In agricultural practice, farmers use two basic Mg fertilization systems, differing in the method of Mg fertilizer application to crops. The first, a classic method, is the in-soil application of Mg fertilizer using lime (Mg oxide or Mg carbonate) for acidic soils and magnesium sulfate (Kieserite) for soils with an optimum pH [19]. Mg availability to plants is not a problem, provided the content of its available form in the soil is in the medium class, at least [19]. The second method of Mg supply to crop plants is foliar fertilization. Magnesium is widely used in this way throughout the world [20,21]. The key challenge for farmers using soluble Mg fertilizers (i.e., sulfate or chloride) is to determine the (i) rate, (ii) methods, and (iii) time—the plant growth stage of fertilizer application. A well-developed Mg fertilization system should be oriented toward an increase in the efficiency of N_f_. 

The interaction between Mg and N occurs at all levels of a plant’s organization. The importance of N for a plant’s growth and yield results from its presence in key biological molecules such as chlorophyll and the ribulose bisphosphate carboxylase–oxygenase enzyme, simply called Rubisco (RuBP) [22]. The latter is a key N-dependent enzyme, decisive in the survival of life on Earth. Its key function is to capture and then fixate the CO_2_ molecule, which is the basic substrate for the production of elementary sugar compounds [23,24]. The key function of Mg is to maintain Rubisco activation, which results in the stabilization of the net photosynthetic rate, as was demonstrated for wheat by Shao et al. [25]. Crop plants well supplied with Mg since the beginning of their growth increases N uptake, resulting in an increase in its unit productivity [26,27]. For example, Mg concentration in maize leaves during the grain filling period is the critical factor affecting the grain yield of this crop. The adequate nutrition of plants with Mg increases N productivity, in turn decreasing the required N_f_ dose [28,29]. 

Two main questions remain open. The first, and most important, concerns the relationship between N and Mg from the point of view of the impact of Mg on the management of N by arable crops. The N/Mg ratio during the early growth of potato tubers or sugar beets is crucial for the final yield of both crops [20,27,30,31]. The second, more minor one concerns the efficiency of the applied Mg, depending on the method of its application. The main objective of this study was to evaluate (i) the effectiveness of applied Mg depending on the fertilization method and (ii) the relationship between Mg and N uptake by wheat and its use efficiency. A secondary objective was to evaluate Mg fertilization systems including soil and foliar fertilization of wheat fertilized with an optimum dose of N_f_. 

## 2. Results

### 2.1. Magnesium Fertilization Systems and Yield Increment 

The yield gain due to the application of Mg to winter wheat was the result of the interaction between the soil and the foliar treatment (Figure 1). Detailed information on the grain yield and the elements of the yield structure can be found in an article by Grzebisz and Potarzycki [32]. In all years of the study, Mg application, regardless of the Mg fertilization system, increased the grain yield. The recorded increase ranged from approximately 0.58 to 0.74 t ha^−1^. The lower yields were due to unfavorable weather conditions during the spring growing season (2013 and 2015). Nevertheless, the effect of the Mg fertilization system (Mg-FS) on the yield gain in the consecutive years of the study was highly stable (Figure A1). A strong interaction between the Mg and FSs was observed (Figure 1). The increase in yield due to the soil-applied Mg (Mgs) in comparison with the absolute Mg control (the plot treated only with NPK) amounted to approximately 0.52 t ha^−1^. The effect of a single stage of Mg foliar treatment (Mgf), regardless of the growth stage of wheat, was 0.57 t ha^−1^. A double stage of Mg foliar application (at BBCH 30 and repeated at BBCH 49/50) resulted in a yield increase of 0.74 t ha^−1^. The effect of the interaction of both Mg and FSs with the yield gain was dependent on the dose of Mgs. The increase in yield was 0.77 t ha^−1^ for the plot fertilized with 25 kg Mg ha^−1^, but it provided Mg foliar application at BBCH 30. The same level of increase in the yield was recorded on the plot with 50 kg Mg ha^−1^, regardless of the growth stage of wheat treated with Mg. In-soil and double, two-phase foliar feeding of wheat with Mg resulted in a yield gain of 0.92 t ha^−1^.

### 2.2. Magnesium Accumulation and Indices of Efficiency 

Magnesium accumulation in wheat at maturity was significantly dependent on the course of the weather during the growing season (Table 1). The effect of the Mg fertilization system was not significant for grain, but it was significant for wheat residues (straw + chaffs). The total Mg accumulated by wheat at harvest resulted from the interaction between the Mg fertilization systems and years (Figure 2). In 2013, the effect of the Mgs was visible only on the Mgs50 plot. The effect of Mgf was the strongest on the Mgs25 main plot. Plants foliar fertilized at the BBCH 49/50 stage increased Mg accumulation by 2 kg ha^−1^ compared to its uptake on the Mgs control plot. In 2014, the uptake of Mg was, in general, higher than in 2013 and 2015. In the Mgs control plot, the double-stage foliar fertilization of wheat increased the uptake of Mg by 2.0 kg ha^−1^ compared to the Mg absolute control. The effect of the Mgs was only slightly stronger on the Mgs50 than on the Mgs25 main plot. The interaction between both fertilization systems was the strongest on the Mgs50 plot and the double-stage Mgf treatment. Mg extra uptake was 2.9 and 2.5 kg ha^−1^ higher compared to the Mg absolute control and the Mgs50 control, respectively. In 2015, the average Mg accumulation by wheat was 30% and 21% lower compared to 2014 and 2013, respectively. The positive effects of Mgf on the uptake of Mg were observed only on the Mgs plots. The strongest increase in Mg accumulation by wheat in response to Mgf was observed on the Mgs50 plot, reaching the maximum for the double-foliar-treated plants (BBCH 30 + 49/50). The greatest amount of Mg taken up by wheat during the growing season (i.e., slightly above two-thirds) was accumulated in grain as indicated by the Mg harvest index (Mg–HI).

Only two indices of Mg productivity (i.e., the Mg unit accumulation in grain (MgUA–G) and the total Mg unit accumulation (MgUA–T) of the nine), showed a significant but negative relationship with the yield (Table A1). Four of the five indices (Mg–HI, MgUA–G, MgUA–T, and Mg unit productivity–grain (MgUP–G)) showed year-to-year variability, but their response to the interaction Y × Mgs × Mgf was not significant. The only significant dependence was noted for the Mg total unit productivity (MgUP–T). However, this index was negatively correlated with the total Mg accumulation (*r* = –0.78 ***). Moreover, its highest values were recorded in the dry year of 2015, when they were 46% and 28% higher compared to 2013 and 2014, respectively. 

Four classic efficiency indices of the applied nutrients (PFP–Mg, Mg–AE, Mg–R, and Mg–PhE) responded significantly to all studied factors including Y × Mgs × Mgf (Table 1). However, none of them showed a significant relationship with the yield (Table A1). All these indices displayed a steady yearly trend, clearly highlighting much higher values in 2014, the year with the highest yield. A significant effect of the experimental treatments on these indices resulted from the increase in the applied doses of Mg from 2.4 to 56.4 kg ha^−1^. This trend can be described by a power regression model. The general trend in the data obtained is presented in detail for Mg recovery (Mg–R) (Figure A2). Its values, regardless of the year of the study, were the highest for the plots with only foliar-applied Mg. The recovery of the applied Mg for these treatments exceeded 100%. The differences between years were the clearest for the Mgf dose of 2.4 kg ha^−1^, reaching 383% in 2014, 300% in 2013, and 260% in 2015. The developed regression models clearly showed that the yearly pattern, observed for the lowest dose of Mg, was maintained with an increase in the Mg dose (Figure A3). The impact of the Mgs × Mgf interaction on the Mg-R values was observed only on the main Mgs25 plot. 

### 2.3. Nitrogen Accumulation and Indices of Efficiency 

The total amount of N accumulated in winter wheat at maturity (TN) significantly depended on the experimental factors. In fact, they influenced the accumulation of N, but the interaction with years was only noted for wheat residues (Table 2). The amount of N in grain (N_a_G) was the result of independent interactions of the year with Mg fertilization systems (Y × Mgs and Y × Mgf). The first were revealed in 2014 and 2015. In both years, the Mgs caused a significant increase in N_a_G compared with the Mg absolute control. A significant effect of Mgs50 was only observed in the dry year of 2015 (Figure 3). The effect of the Mgf was most evident in 2013 (Figure 4). In 2014, it was only slightly marked at BBCH 30. In 2015, it was much more visible, but a significant difference was only noted between the Mgs control and the plot with double Mg foliar fertilization (BBCH 30 + 49/50). The same principles were noted for the nitrogen harvest index (NHI). The values of this index were very high, approaching almost 90% in 2014.

Of the nine nitrogen use efficiency (NUE) indices studied, only three (i.e., nitrogen unit accumulation in grain (NUA–G), nitrogen unit productivity for grain (NUP–G), and total nitrogen unit productivity (NUP–T)) responded significantly to the Y × Mgs × Mgf interaction. The NUP–G was ultimately the most sensitive index (Figure 5). In 2013, its values showed no response to experimental factors, oscillating around 46 ± 0.8 kg grain kg^‒1^ Na. In 2015, the NUP–G was significantly lower, on average, reaching only 37 ± 1.5. In 2014, the highest N productivity was found for the Mgs control. The highest NUP–G of 50 kg grain kg^‒1^ Na was recorded on the plot with Mgf at the BBCH 49/50 stage. On plots with soil-applied Mg, the NUP‒G indices were slightly lower than for the Mgs control but at the same time significantly responded to the Mgf treatments. Despite that, the NUP–G did not show a significant impact on the yield (Table A2). Moreover, it was negatively correlated with Na–G, Na–CR, TN, and, as a rule, with NUA–G and NUA–T, and also with N–R. At the same time, this particular index was significantly and positively correlated with N‒PhE (r = 0.98 ***).

The classic NUE index, the partial factor productivity of fertilizer N (PFP–N_f_), responded to the Mgs × Mgf interaction, and the nitrogen physiological efficiency (N-PhE) responded to the Y × Mgf interaction. The application of Mg significantly increased the productivity of N_f_. The differences between the Mg absolute control plot and plots fertilized with Mg were significant. The strongest increase in PFP-N_f_ was recorded for Mgs50, which provided double-foliar fertilization with Mg. The highest variability in N–PhE indices was recorded in 2013 (Figure 6). Nitrogen utilization by wheat increased in response to Mg foliar fertilization, peaking when applied at BBCH 49/50. In the remaining two years, N–PhE showed significantly lower variability in response to Mgf, especially on plots with double Mg treatments. Two other NUE indices (i.e., nitrogen apparent efficiency (NAE) and nitrogen recovery (N–R)) responded significantly to the studied factors but did not interact with each other. The NAE indices were highly sensitive to the weather, having the highest values in 2013 (31% higher than in the dry year of 2015). A reverse trend was recorded for N–R, which was high in all years but exceeded 100% in 2015. The soil-applied Mg increased the values of both indices. The same effect was observed with Mg foliar application. 

The wheat yield showed the highest correlation with N–HI and then with NAE (Table A2). The yield formula based on NHI, despite a statistically proven linear model, can be represented as a quadratic function:GY = 0.26NHI − 11.54 for *n* = 36, R^2^ = 0.82 and *p* ≤ 0.01 (t ha^−1^)(1)
GY = 0.016NHI^2^ + 2.9NHI − 122.7 for *n* = 36, R^2^ = 0.85, *p* ≤ 0.01 (t ha^−1^)(2)

The second equation indicates that an N–HI of 92.4% would give a yield of 11.22 t ha^−1^. The maximum yield of 11.18 t ha^−1^ was obtained when the NHI reached 88%. This effect was recorded in 2014 on the Mgs25 plot and Mgf at BBCH 49/50. N–HI was negatively correlated with the amount of N in crop residues and with NUA–G and NUA–T. The highest positive relationships were recorded with PFP–N (*r* = 0.91 ***) and NAE (*r* = 0.81 ***). 

Magnesium fertilization, as previously documented, significantly affected its accumulation in wheat, but its total accumulation (Mg_a_T) was sensitive to the interaction of experimental factors with years (Table A3). The most significant impact of Mg_a_T was recorded for N_a_CR:N_a_CR = −4.16Mg_a_T + 111.4 for *n* = 36, R^2^ = 0.70 and *p* ≤ 0.01 (kg ha^−1^)(3)

The increase in Mg_a_T decreased the N accumulated in crop residues. The response of the NUE indices to Mg_a_T, as well as to Mg_a_G as a major part of Mg_a_T, was very specific. A decrease was noted for NUA–G, NUA–T, and N-R (Table A3). An increase was recorded for NUP–G, NUP–T, and N–PhE. PFP–N and NAE were significantly correlated with each other and deserve special attention. PFP–N responded positively but weakly to the amount of Mg_a_G or Mg_a_T. In contrast, NAE showed a highly positive, linear response to the Mg_a_T (Figure A4):NAE = 1.34Mg_a_T + 8.93 for *n* = 36, R^2^ = 0.67 and *p* ≤ 0.01 (grain kg kg^−1^ Nf)(4)

At the same time, NAE was significantly correlated with both NUP indices but especially with NUP–T (*r* = 0.84 ***) (Table A2). 

## 3. Discussion

### 3.1. Magnesium Use Efficiency

An increase in the yields of crop plants in response to the applied nutrient, regardless of the method of application, is the essence of the application of any fertilizer [33]. An assessment of wheat’s response to Mg fertilization requires answers to three basic questions: Is there an advantage of soil over foliar Mg fertilization?Does the stage of wheat development affect the choice of the date for foliar Mg fertilization?Is there any interaction between the two fertilization systems with regard to the end result, i.e., yield increase?

The net increase in the yield of winter wheat due to the different systems of Mg fertilization was significant, regardless of the course of the weather over the studied years. In general, the increase in the yield resulting from the single-stage foliar application of Mg was only slightly higher than that of soil-applied Mg (0.52 vs. 0.57 t ha^−1^). The increase in wheat yield as a result of Mg foliar fertilization increased up to 0.7 t ha^−1^ but provided its double application at BBCH 30 and repeated at BBCH 49/50. The end of booting/beginning of heading is considered the optimal stage for wheat foliar Mg fertilization [26,27]. This stage is crucial for the number of grains per unit area (GD) [34]. In the presented case, the recorded increase in the yield resulted directly from the increase in the GD [32]. The best effect of Mg application on wheat, resulting in a yield gain of 0.92 t ha^−1^, was due to the interaction of both Mg fertilization systems. The course of weather during the growing season did not markedly change the trend of the wheat response to the tested Mg fertilization systems. On this basis and according to the data in the literature, three strategies for wheat fertilization with Mg during growth can be identified:Conservative: a high and stable yield increase, increasing the resistance to abiotic stresses;Effective: a moderate-to-high yield increase, provided there are no abiotic stresses;Prophylactic: a moderate yield increase, with a constant fertilization factor.

The first strategy was very efficient in years with some disturbances in the weather course during the spring growing season of winter wheat (Figure 1). It requires, however, a higher dose of the in-soil applied Mg and its frequent application to wheat foliage [14,19,20]. The yield-forming effect of the applied Mg can be explained both by the increase in the GD and the maintenance of the photosynthetic activity of leaves during the grain-filling period [24,25,32]. The second Mg fertilization strategy can be recommended in the years or regions of the world with favorable growth conditions for winter wheat. A sufficiently high yield increase can be achieved by applying relatively low in-soil Mg doses, provided a high solubility of fertilizer is used as well as one-stage foliar fertilization with Mg [19,26]. The third fertilization strategy of winter wheat with Mg is based on a double-stage foliar fertilization, regardless of the weather and soil conditions [27]. 

Mg foliar fertilization was superior to its soil application, but provided a double-stage application, as supported by the obtained data from its extreme efficiency. The values of the Mg recovery indices (Mg–R) on the main Mgf plot ranged from 100% to 300%. In contrast, the Mg recovery on the Mgs plots were in the range of 18–38% for the Mgs25, and 10–19% for the Mgs50 main plots. Moreover, the highest Mg–R indices, regardless of the method of Mg application, were recorded in 2014, the year with the highest grain yield. The extremely high Mg–R values of the Mgf plot were due to the higher Mg uptake by plants and its accumulation in vegetative wheat biomass. The increase in Mg uptake, averaged over treatments, was 16.6% higher compared to the absolute Mg control. It is well documented in the scientific literature that chlorophyll Mg, despite being a stable plant trait, is the main source of N for the growing seed/grain [35]. The obtained results suggest an extension of the grain-filling period (GFP), which resulted in the higher yield. This conclusion is confirmed by Ahnadi-Lahijani and Emam [36], who showed that the higher the chlorophyll content in wheat leaves during GFP and the longer the leaf surface is kept green, the greater the increase in wheat yield. 

### 3.2. Impact of Magnesium Uptake on Nitrogen Management by Wheat 

The fourth question concerns the impact of the Mg fertilization system on N management, i.e., N uptake by wheat and its net utilization. The conducted study clearly showed that wheat treated with Mg, regardless of the method of its application, significantly increased the amount of N in wheat at harvest (TN). It should be emphasized that the N taken up by wheat in response to Mg application was mainly accumulated in the grain (Table 2). Both experimental factors contributed to this but without interaction between the systems. The soil-applied Mg increased the amount of N in wheat grain by 24 and 21 kg ha^−1^ in 2014 and 2015. Assuming a protein concentration in the grain at a level of 13%, the corresponding yield increase would be at a level of 1.1 and 0.93 t ha^−1^. These values are almost equal to the yield increment as a result of the interaction of both mg fertilization systems (Figure 1). Thus, it can be concluded that regardless of weather conditions, the increase in yield is directly related to the efficiency of N utilized by wheat. 

The strong relationship between the amount of extra N accumulated in wheat grain, as a result of the of Mg fertilization, was due to the critical role of both nutrients in photosynthesis. Nitrogen is the limiting component of Rubisco, the key plant enzyme responsible for CO_2_ fixation [24,25]. The influence of Mg on the activity of Rubisco increases in conditions of water shortage, which is often accompanied by elevated temperatures [36,37]. Rubisco activity increases during the GFP of wheat growth, as recently documented by Shao et al. [25]. This specific effect was probably observed in 2015 on the main plot fertilized with 50 kg Mg ha^−1^ and double Mg fertilization. The stabilizing effect of the soil-applied Mg on the yield of crops is well documented for various biologically different plants such as maize and sugar beets [28,29,30,31]. The assumed stabilization results from a plant’s accessibility to the readily available Mg, even from the beginning of growth [19,26]. In wheat, the critical period of yield formation starts with the beginning of the elongation phase [8,9]. However, *the critical window* for wheat grain density takes place during the booting and heading stages [8,38]. Foliar Mg fertilization at these stages results in a higher grain density in cereals and maize [27]. 

Out of the nine studied NUE indices, only four were significantly related to the yield of wheat. The highest relationship was found for the nitrogen harvest index (N–HI). This index is a rule treated as a conservative wheat trait [39]. The study clearly showed its significant dependence on the total amount of Mg in wheat at harvest. The effect of Mg on N–HI was indirect, i.e., it reduced the amount of N in wheat residues. This trend means a greater transfer of N during GFP from the vegetative parts of wheat to the growing grains. The proposed explanation confirms an earlier study by Potarzycki [40,41]. The author showed that foliar Mg applied to wheat at the beginning of the booting phase increased the remobilization of N from vegetative tissues to grain. 

Nitrogen apparent efficiency (NAE) was the second-most important (*p* ≤ 0.01) NUE index, showing a significant relationship with the yield. It also depended on the total amount of Mg in the wheat biomass or grain (Figure A4). The positive relationship between this index and wheat grain yield indirectly indicates a higher yield from plots fertilized with Mg. Moreover, this index was inversely correlated with the amount of N accumulated in wheat residues but positively with N–HI, NUP–G, NUP–T and, finally, N–PhE. On the basis of the obtained relationships, it can be concluded that, regardless of the fertilization system, Mg had a strong, significant effect on N utilization by winter wheat (Figure A4). The positive effect of Mg on the productivity of N_f_ was confirmed by the increased productivity of applied fertilizer N, as confirmed by the response of the PFP–N_f_ index. Its values ranged from 45 kg grain kg^−1^ N_f_ in 2013, an unfavorable year for wheat growth, to 58 kg grain kg^−1^ N_f_ in 2014, the year with the highest yields. The highest PFP-N_f_ values for an N_f_ rate of 190 kg N ha^−1^ are comparable to those presented by Szczepaniak et al. [42] for wheat fertilized with 160 kg N ha^−1^. The effective use of N_f_ by winter wheat through the use of Mg is emphasized by the values of the nitrogen recovery index (N–R). It was very high on the Mg absolute control plot (NPK only), well above 80%. The use of Mg raised its values above 90%. Moreover, the impact of Mg on this index was very stable, regardless of the weather conditions during the growing season and the method of Mg application.

## 4. Materials and Methods

### 4.1. Experimental Site 

A field experiment was carried out at Jarosławiec (52°15′ N, 17°32′ E, Poland) on soil originated from sandy loam, classified as Albic Luvisols (Neocambic) [43]. The content of organic matter (C_org_) in a 0.0–0.3 m layer during the study ranged from 21 ± 0.1 to 25 ± 0.9 g kg^−1^ soil (losses on ignition). Soil reaction (pH) was in the neutral range (1 M KCl). The content of available nutrients, measured before the application of fertilizers was, in general, good for P and sufficient for K and Mg. The amount of the mineral N (N_min_), determined in a 0.0–0.9 m layer, was high in the first two growing seasons and medium in the third (Table 3).

### 4.2. Weather Conditions

The weather conditions were very variable in the consecutive growing seasons (Figure 7). The beginning of spring, with the exception of 2012/2013, favored the growth of wheat. In 2013, negative temperatures in the first two decades of March arrested plant growth. In all years of the study, temperatures during flowering and grain filling were within the ranges optimal for yield development. The sum of rainfall during the spring growing season was as follows: 2013—299.4, 2014—285.2, and 2015—265 mm. In 2015, a shortage of rainfall was revealed, which covered three main phases of wheat development, ranging from shooting to early flowering. The sum of rainfall in this period was 37.6 mm, while in 2013, it reached 88.8 mm, and in 2014, 92.6 mm.

### 4.3. Experimental Design

The field experiment was arranged as a two-factor split-plot design, replicated 4-fold: Soil-applied magnesium (Mgs): 0, 25, and 50 kg Mg ha^−1^ (acronym: Mg control, Mgs25, and Mgs50);Foliar-applied magnesium (Mgf):Without application, i.e., Mgf control;Applied at the BBCH 30 stage (I) (I–BBCH 30);Applied at the BBCH 49/50 stage (II) (II–BBCH 59/50);Applied at the BBCH 30/31 and BBCH 49/50 stages; double-stage application (I + II). 

Spring barley was the fore-crop for winter wheat. The *Tobak* wheat variety was sown annually on 20–25 September. The soil was fertilized with Mg in the form of Kieserite (MgSO_4_ · H_2_O), containing 25% MgO and 50% SO_3_. Kieserite was applied to the soil three weeks before wheat sowing. Foliar fertilization of wheat with Mg was carried out using Epsom salt (MgSO_4_ · H_2_O) containing 16% MgO and 37.5% SO_3_. The amounts of the applied nutrients are shown in Table 4. Sulfur applied together with the Mg fertilizer was balanced in the first dose of N. It was used as a mixture of ammonium sulfate and ammonium saltpeter (17.5% SO_3_). The first N dose of 80 kg ha^−1^ was applied just before the beginning of the growing season in spring. The second dose of 50 kg ha^−1^ was applied, and the third one of 60 kg ha^−1^ at BBCH 45–47. Phosphorus at a rate of 30.1 kg P ha^−1^ as triple superphosphate (46% P_2_O_5_) and K at a rate of 66.4 kg K ha^−1^ as muriate of potash (KCl) were applied together with the soil Mg. The total area of a single plot was 30 m^2^, and the harvested area was 15 m^2^. Plant protection was conducted in accordance with the codex of good practice.

### 4.4. Plant Material Sampling and Analysis

The plant material used for dry matter determination was collected at BBCH 89 from an area of 2.0 m^2^. The sampled material was then divided, depending on the wheat stage, into subsamples of grain (G) and crop residues (CRs) composed of leaves (LE), stems (ST), ears (EA), and chaffs (CH). The results are expressed on a dry weight basis. 

The N content was determined in both parts of the plant, using the standard macro–Kjeldahl procedure. For determination of the Mg content, the plant sample was dried at 65 °C and then mineralized at 550 °C. The obtained ash was then dissolved in 33% HNO_3_. The concentration of Mg was determined using atomic absorption spectrometry—flame type. The results are expressed on a dry matter basis.

### 4.5. Parameters and Indices of Nitrogen Use Efficiency

The equations used to calculate the amount of N in the grain or crop residues and the N use efficiency (NUE) indices are presented below. The corresponding Mg indices were calculated in the same way. 

Nitrogen accumulation in wheat grain, *NaG:*$$N_{a}G=GY\times N\;\;\text{kg}\;{\text{ha}}^{-1}$$Nitrogen accumulation in crop residues, *N_r_:*$$N_{a}CR_{s}=CR_{s}\times N\;\;\text{kg}\;{\text{ha}}^{-1}$$Total accumulation of nitrogen in wheat biomass, TN:$$TN=N_{a}G+N_{a}CR_{s\;}\text{kg}\;{\text{ha}}^{-1}$$Nitrogen harvest index, *NHI:*$$NHI=\frac{N_{a}G}{TN}\times 100\%$$Nitrogen unit accumulation in grain, *NUA-G:*$$NUA=\frac{N_{a}G}{GY}\;\;\text{kg}\;N\times t^{-1}$$Nitrogen unit accumulation in total wheat biomass, *NUA-T:*$$NUA=\frac{TN}{GY}\;\;\text{kg}\;N\times t^{-1}$$Nitrogen unit productivity—grain, *NUP*−*G:*
$$NUP-G=\frac{GY\times 1000}{N_{a}G\;}\;\;\text{kg}\;grain\times {\text{kg}}^{-1}\;N$$Nitrogen unit productivity–total, *NUP-T:*
$$NUP-T=\frac{GY\times 1000}{TN}\;\;\text{kg}\;grain\times {\text{kg}}^{-1}\;N$$Partial factor productivity of fertilizer *N*, *PFP-N:*$$PFP-N=\frac{GY\times 1000}{N_{f}}\;\;\text{kg}\;grain\;{\text{kg}}^{-1}\;N_{f}$$Nitrogen agronomic efficiency, *NAE*:$$NAE=\frac{GY_{f}\times 1000-GY_{N}{}_{c}\times 1000}{N_{if}}\;\;\text{kg}\;grain\times {\text{kg}}^{-1}\;\;N_{f}$$Nitrogen recovery, *N–R:*
$$N-R=\frac{TN_{N_{f}}-TN_{N_{c}}}{N_{f}}\times 100\%$$Nitrogen physiological efficiency, *N-PhE:*
$$N-PhE=\frac{GY_{f}\times 1000-GY_{N}{}_{c}\times 1000}{TN_{N_{f}}-TN_{N_{c}}}\;\;\text{kg}\;grain\;{\text{kg}}^{-1}\;N$$
where:

*NHI*—nitrogen harvest index, %;

*CRs*—crop residues, t ha^−1^;

*N*—*N* content in grain or crop residues, %, g kg^−1^ DW; 

*N_f_*—plots fertilized with N;

*N_c_*—nitrogen control;

*N_f_*—treatment fertilized with nitrogen. 

### 4.6. Statistical Analysis 

The collected data were subjected to an analysis of variance using STATISTICA^®^ 13 (StatSoft, Inc., Krakow, Poland, 2013). The distribution of the data (normality) was checked using the Shapiro–Wilk test. The homogeneity of variance was checked by the Bartlett test. Means were separated by honest significant difference (HSD) using Tukey’s method, where the *F*-test indicated significant factorial effects at a level of *p* < 0.05. To determine the wheat grain yield, stepwise regression was used to define the optimal set of wheat components. In the computational procedure, a consecutive variable was removed from the multiple regressions in a step-by-step manner. The best regression model was chosen based on the highest F–value for the model and the significance of all variables. 

## 5. Conclusions

Magnesium applied to wheat resulted in a significant yield gain with respect to the effect of NPK, treated as the Mg control. The method of application was of secondary importance. A slightly higher increase in the yield was caused by foliar fertilization, preferably performed at the booting/heading stages of wheat growth. The yield gain, as a result of foliar fertilization with Mg fertilization, ranged from 0.6 to 0.9 t ha^−1^, while in the soil, its application resulted in a yield gain in the range of 0.4–0.7 t ha^−1^. Magnesium accumulation by wheat, averaged for the fertilization treatments, increased by 17% compared to the NPK plot. The recovery of foliar-applied Mg was multiple in relation to the applied dose. The recovery of soil-applied Mg depended on the dose, ranging from 18 to 38% on the 25 kg Mg ha^−1^ main plot and from 10 to 19% on the 50 kg Mg ha^−1^ plot. The main effect of wheat fertilization with Mg was its impact on the uptake and then partitioning of the accumulated N in wheat biomass between the grain and crop residues. The amount of extra accumulated N was effectively converted into grain yield. This process manifested itself in an increase in the value of the nitrogen harvest index and in a decrease in the N content in crop residues. 

The main action of Mg, regardless of the weather and the method of its application, was an increase in the productivity of fertilizer nitrogen, which was confirmed by a set of various tested indices such as NUP–G, NUP–T, N–PhE, and PFP–N_f_. The yield-forming effect of the applied Mg fertilizer to winter wheat was revealed by the increased N transfer to the grain, which indicates its impact on the nitrogen utilization efficiency. 

## Figures and Tables

**Figure 1 plants-11-02600-f001:**
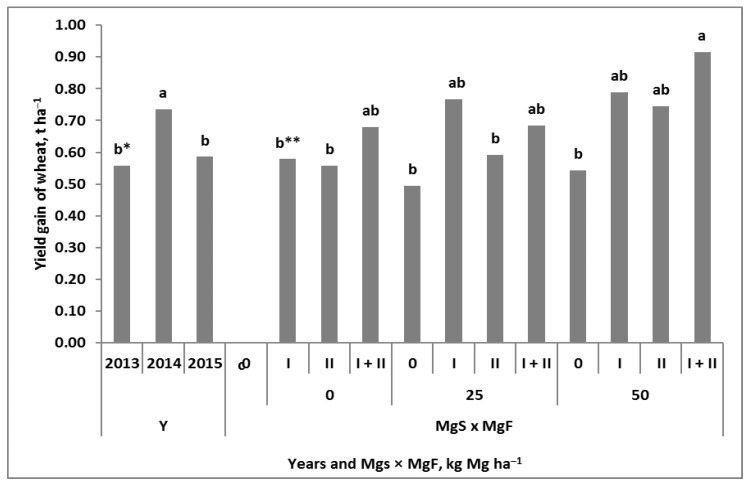
The net increase in the yield of winter wheat in response to the interaction between Mg fertilization systems. a, b. Similar letters mean a lack of significant differences using Tukey’s test. Mgs and Mgf—soil and foliar Mg fertilization systems, respectively. * Doses of applied Mg, kg ha^−1^; ** stages of Mg. Wheat stages of Mg foliar fertilization: I—BBCH 30; II—BBCH 49/50.

**Figure 2 plants-11-02600-f002:**
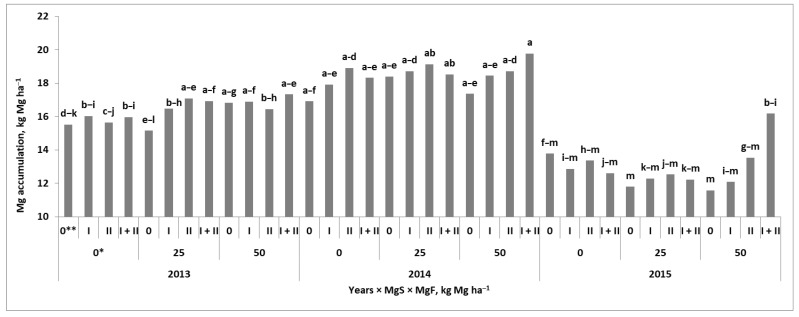
Magnesium accumulation by winter wheat in response to Mg fertilization systems and years. a, b, c, d, e, f, g, h, i, j, k, l, m Similar letters mean a lack of significance; differences using Tukey’s test. Mgs and Mgf—soil and foliar Mg fertilization systems, respectively. * Doses of applied Mg, kg ha^−1^; ** stages of Mg foliar fertilization to wheat: I—BBCH 30; II—BBCH 49/50.

**Figure 3 plants-11-02600-f003:**
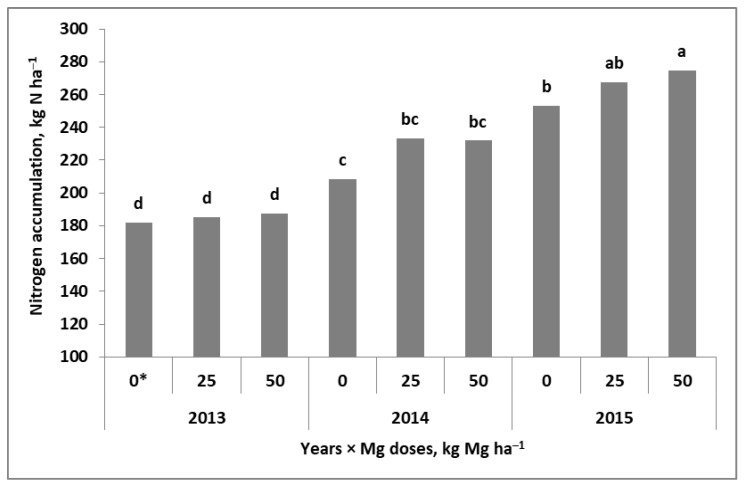
Effect of soil-applied magnesium to winter wheat on nitrogen accumulation in grain. a, b, c, d Similar letters mean a lack of significant differences using Tukey’s test. Mgs—soil Mg fertilization system. * Doses of applied Mg, kg ha^−1^.

**Figure 4 plants-11-02600-f004:**
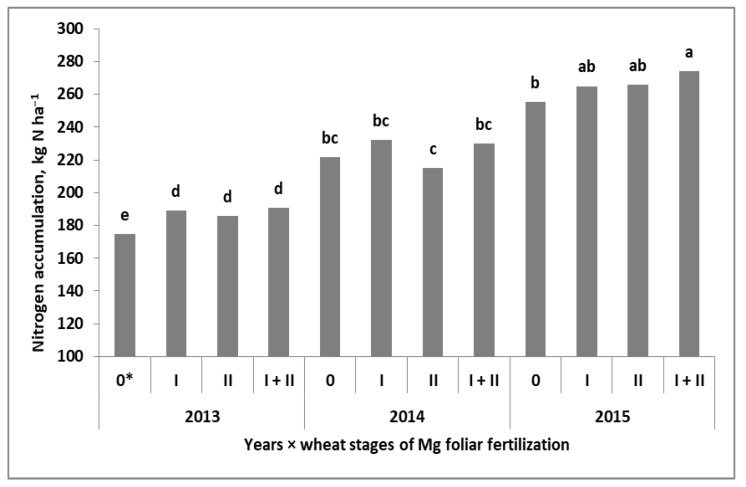
Effect of foliar fertilization of winter wheat with magnesium on nitrogen accumulation in grain. a, b, c, d, e Similar letters mean a lack of significant differences using Tukey’s test. Mgf—foliar Mg fertilization system. * Stages of Mg foliar fertilization to wheat: I—BBCH 30; II—BBCH 49/50.

**Figure 5 plants-11-02600-f005:**
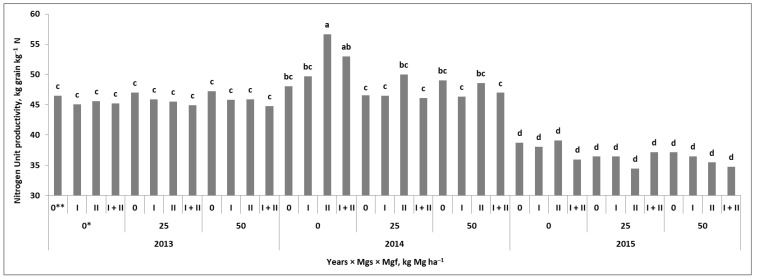
The nitrogen unit productivity of grain in response to the interaction of Mg fertilization systems and years. a, b, c, d Similar letters mean a lack of significant differences using Tukey’s test. Mgs and Mgf—soil and foliar Mg fertilization system, respectively. * Doses of applied Mg, kg ha^−1^; ** stages of Mg foliar fertilization to wheat: IBBCH—30; II—BBCH 49/50.

**Figure 6 plants-11-02600-f006:**
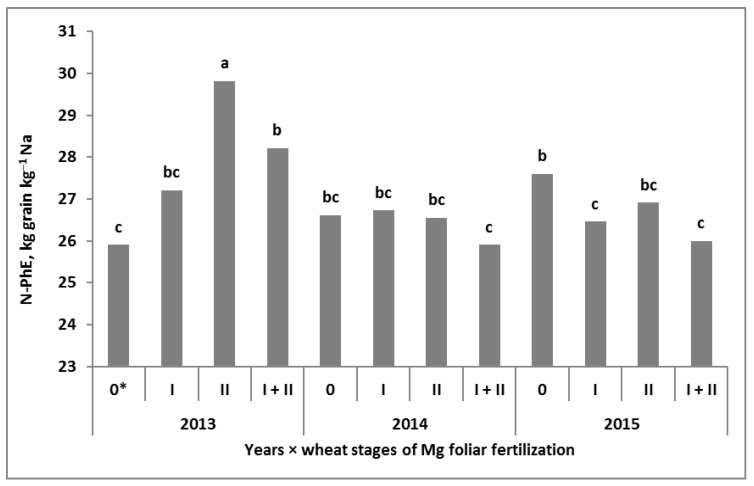
Effect of winter wheat foliar fertilization with magnesium on nitrogen physiological efficiency. a, b, c Similar letters mean a lack of significant differences using Tukey’s test. Mgf—foliar Mg fertilization system. * Stages of Mg foliar fertilization to wheat: I—BBCH30; II—BBCH 49/50.

**Figure 7 plants-11-02600-f007:**
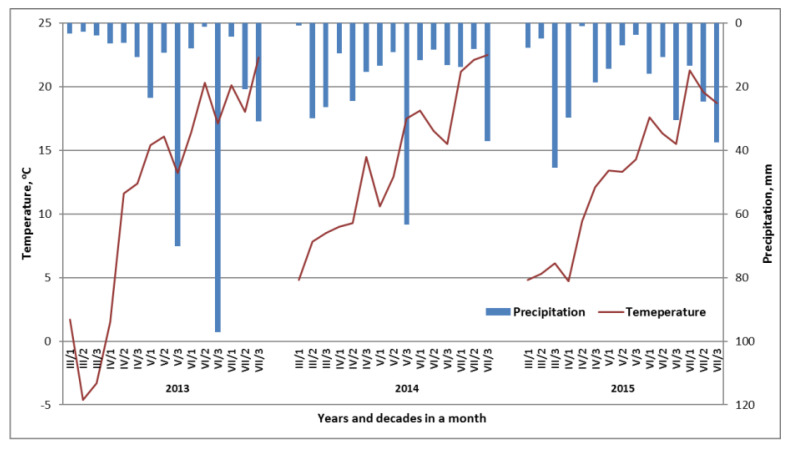
Weather conditions during the consecutive growing seasons.

**Table 1 plants-11-02600-t001:** Magnesium uptake by winter wheat at maturity and indices of magnesium use efficiency.

Factor	Level of Factor	Mg_a_G	Mg_a_CR	MgaT	MgHI	MgUA–G	MgUA–T	MgUP–G	MgUP–T	PFP–Mg	MgAE	Mg–R	Mg–PhE
		kg ha^−1^			%	kg Mg t^−1^		kg Grain kg^−1^ Mg			t Grain kg^−1^ Mgf	%	t Grain kg^−1^ Mg_a_T
Year	2013	11.1 b	5.3 b	16.4 b	67.9 a	1.31 a	1.93 a	766.2 c	519.9 c	731.8 c	0.36 b	62.6 b	0.66 b
(Y)	2014	12.6 a	5.8 a	18.4 a	68.4 a	1.15 b	1.69 b	871.6 b	595.4 b	946.8 a	0.44 a	81.6 a	1.05 a
	2015	8.3 c	4.6 c	12.9 c	64.8 b	0.86 c	1.33 c	1178.8 a	760.5 a	843.3 b	0.34 b	54.4 c	0.56 b
*p*		***	***	***	***	***	***	***	***	***	***	***	***
Mg in-soil	0	10.6	5.1 a	15.6 b	67.4	1.11	1.65	921.5	620.1	1990.1 a	0.91 a	156.2 a	2.21 a
Mgs	25	10.7	5.1 a	15.8 ab	67.3	1.10	1.63	950.7	634.6	346.8 b	0.16 b	27.2 b	0.04 b
	50	10.8	5.5 a	16.3 a	66.3	1.11	1.67	944.3	621.2	185.0 c	0.09 c	15.2 c	0.01 c
*p*		n.s.	*	*	n.s.	n.s.	n.s.	n.s.	n.s.	***	***	***	***
Mg foliar	0	10.4	4.8 b	15.2 b	68.3 a	1.11	1.63	929.0	633.2	191.9 d	0.09 d	14.0 d	0.02 d
(Mgs)	I	10.6	5.1 ab	15.7 ab	67.1 ab	1.09	1.62	959.9	638.5	1524.4 a	0.69 a	118.1 a	1.96 a
	II	10.9	5.3 ab	16.1 a	67.1 ab	1.12	1.67	919.5	614.2	975.2 b	0.44 b	79.7 b	0.75 b
	I + II	10.8	5.6 a	16.4 a	65.7 b	1.10	1.68	947.0	615.3	671.2 c	0.31 c	53.0 c	0.30 c
*p*		n.s.	***	***	*	n.s.	n.s.	n.s.	n.s.	***	***	***	***
Source of Variation for Interaction													
Y × Mgs		*	n.s.	*	*	*	n.s.	*	*	***	***	***	***
Y × Mgf		n.s.	n.s.	n.s.	n.s.	n.s.	*	n.s.	n.s.	***	***	**	***
Mgs × Mgf		n.s.	**	**	*	n.s.	n.s.	n.s.	***	***	***	***	***
Y × Mgs × Mgf		n.s.	*	*	n.s.	n.s.	n.s.	n.s.	**	***	***	***	***

a, b, c, d Similar letters mean a lack of significant differences using Tukey’s test; ***, **, and * indicate significant differences at *p* < 0.001, *p* < 0.01, and *p* < 0.05, respectively; 30% n.s.—not significant. ^1^ I—BBCH 30; II—BBCH 49/50. Mgf—Mg fertilizer doses; Mg_a_G, Mg_a_CR, and Mg_a_T—magnesium accumulation: grain, crop residues, and total, respectively; MgHI—magnesium harvest index; MgUA–G and MgUA–T—magnesium accumulation: grain and total, respectively; MgUP–G and MgUP–T—magnesium unit productivity: grain and total, respectively; PFP–Mg—partial factor productivity of fertilizer magnesium; MgAE—magnesium apparent efficiency; Mg–R—magnesium recovery; Mg–PhE—magnesium physiological efficiency.

**Table 2 plants-11-02600-t002:** Nitrogen uptake by winter wheat at maturity and indices of nitrogen use efficiency.

Factor	Level of Factor	N_a_G	N_a_CR	TN	NHI	NUA–G	NUA–T	NUP–G	NUP–T	PFP–N	NAE	N–R	N–PhE
		kg N ha^−1^			%	kg N t^−1^		kg Grain kg^−1^ N			Grain kg^−1^ N_f_	%	Grain kg^−1^ TN
Year	2013	185.0 c	46.8 b	231.8 c	79.8 c	21.9 b	27.4 b	45.8 b	36.5 b	44.5	22.1 b	78.9 b	28.0 b
(Y)	2014	224.5 b	30.3 c	254.8 b	88.1 a	20.5 c	23.3 c	49.0 a	43.1 a	57.5	27.1 a	81.5 b	33.6 a
	2015	265.0 a	59.0 a	324.0 a	81.8 b	27.3 a	33.4 a	36.7 c	30.0 c	51.0	20.7 c	107.2 a	19.4 c
*p*		***	***	***	***	***	***	***	***	***	***	***	***
Mg in-soil	0	214.5 b	45.1	259.6 b	82.6	22.6 b	27.5 b	45.1 a	37.4 a	50.2 b	22.4 b	83.6 b	27.8 a
Mgs	25	228.8 a	44.9	273.7 a	83.6	23.6 a	28.3 a	43.1 b	36.1 b	51.1 a	23.4 ab	91.0 a	26.4 a
	50	231.2 a	46.0	277.2 a	83.5	23.5 a	28.3 a	43.2 b	36.2 b	51.7 a	24.0 a	92.9 a	26.7 a
*p*		***	n.s.	***	n.s.	***	**	***	***	***	**	***	n.s.
Mg foliar	0	217.1 b	43.6	260.6 b	83.2	23.0	27.8	44.1	36.7	49.6 b	21.9 b	84.2 b	26.7
(Mgf)	I	228.6 a	45.5	274.2 a	83.5	23.4	28.1	43.4	36.3	51.5 a	23.8 a	91.3 a	26.8
	II	222.1 ab	46.1	268.1 ab	82.9	23.0	27.9	44.6	37.1	51.1 a	23.4 a	88.1 ab	27.8
	I + II	231.5 a	46.2	277.7 a	83.4	23.6	28.4	43.2	36.1	51.8 a	24.1 a	93.2 a	26.7
*p*		***	n.s.	***	n.s.	n.s.	n.s.	n.s.	n.s.	***	***	***	n.s.
Source of Variation for Interaction													
Y × Mgs		**	**	*	**	**	n.s.	***	**	n.s.	n.s.	n.s.	n.s.
Y × Mgf		*	*	n.s.	*	***	*	***	**	n.s.	n.s.	n.s.	*
Mgs × Mgf		n.s.	n.s.	n.s.	n.s.	n.s.	n.s.	*	n.s.	*	n.s.	n.s.	n.s.
Y × Mgs × Mgf		n.s.	*	n.s.	n.s.	*	n.s.	*	*	n.s.	n.s.	n.s.	n.s.

a, b, c Similar letters mean a lack of significant differences using Tukey’s test. ***, **, and * indicate significant differences at *p* < 0.001, *p* < 0.01, and *p* < 0.05, respectively; n.s. = not significant. I—BBCH 30; II—BBCH 49/50. N_f_–nitrogen dose; N_a_G, N_a_CR, and N_a_T—nitrogen accumulation: grain, crop residues, and total, respectively; NHI—nitrogen harvest index; NUA–G, NUA–T—nitrogen accumulation: grain and total, respectively; NUP–G and NUP–T—nitrogen unit productivity: grain and total, respectively; PFP–N—partial factor productivity of fertilizer nitrogen; NAE—nitrogen apparent efficiency; N–R—nitrogen recovery; N–PhE—nitrogen physiological efficiency.

**Table 3 plants-11-02600-t003:** Soil characteristics of the experimental plots during the 2012–2015 growing seasons.

Year	Soil Layer	pH	P ^1^	K ^1^	Mg ^2^	N_min_
	(cm)	1 M KCl	mg kg^−1^ Soil			kg ha^−1^
2012/2013	0–30	6.5	63.2 ^1^ M ^3^	157.7 M	33.2 H	76 ^4^
	30–60		59.7 ^1^ M	107.9 M	25.3 M	
2013/2014	0–30	6.7	91.6 ^1^ H	168.0 M	30.2 M	74
	30–60		91.6 ^1^ H	153.6 M	24.7 M	
2014/2015	0–30	6.6	87.2 ^1^ H	182.6 H	24.1 M	57
	30–60		95.9 ^1^ H	149.9 M	30.2 M	

^1^ Egner–Riehm method; ^2^ Schachtschabel method; ^3^ classes of the available nutrient content: M—medium, H—high; ^4^ layer: 0–90 cm (measured in 0.01 M CaCl_2_).

**Table 4 plants-11-02600-t004:** Fertilization schedule.

Treatment	Fertilization Schedule	N–P_2_O_5_–K_2_O	Mg–Soil	Mg–Foliar
		kg ha^−1^		
1.1	NPK	190–70–80	0	0
2.1	NPK–Mg foliar BBCH 30	190–70–80	0	2.4
2.2	NPK–Mg foliar BBCH 49/50	190–70–80	0	4.0
2.3	NPK–Mg foliar BBCH 30 + 49.50	190–70–80	0	6.4
3.1	NPK–Mg soil	190–70–80	25	0
3.2	NPK–Mg soil + foliar BBCH 30	190–70–80	25	2.4
3.3	NPK–Mg soil + foliar BBCH 49/50	190–70–80	25	4.0
3.4	NPK–Mg soil + foliar BBCH 30 + 49/50	190–70–80	25	6.4
4.1	NPK–Mg soil	190–70–80	50	0
4.2	NPK–Mg soil + foliar BBCH 30	190–70–80	50	2.4
4.3	NPK–Mg soil + foliar BBCH 49/50	190–70–80	50	4.0
4.4	NPK–Mg soil + foliar BBCH 30 + 49/50	190–70–80	50	6.4

## Data Availability

Not applicable.

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
