# Peer review of "Magnesium Fertilization Increases Nitrogen Use Efficiency in Winter Wheat (*Triticum aestivum* L.)"

_plants, 2022, doi:10.3390/plants11192600_

Round 1
Reviewer 1 Report
This was interesting manuscript and I enjoyed learning about this topic and the methods used. The paper reads OK and the information presented is relevant to expand the knowledge base/expand the volume of science. I liked the way the tables presented the data, but the figures do not make sense, since the comparison is across all site-years. This makes it difficult to really see the treatment effects, since they differed by year. Table 2 should include comparisons of same Mg rates applied to the soil vs applied foliar, for each Mg rates. Looking at the supplemental materials (fertilization schedule table) - not clear how these NPK rates were decided upon -are these typical rates for the area/wheat class? I might have missed this. Some rewording might be good throughout - I do recommend an English native speaker or an editor to look the text over. For example: in Conclusion section it is worded as :"Magnesium applied to wheat resulted in a significant yield gain with respect to the effect of NPK. The method of application was of secondary importance." In this case, I would emphasize that the RATE of Mg had a more substantial impact on wheat yield, compared to Mg application method. Also, in statistical methods: "To determine the wheat grain yield, stepwise regression was used to define the optimal set of wheat components."- This statement does not read right, or is part of the statement missing?
Author Response
Review report 1_response
I liked the way the tables presented the data, but the figures do not make sense, since the comparison is across all site-years. This makes it difficult to really see the treatment effects, since they differed by year.
Response
Of course, it would be better if the farmer had a clear answer about the practical value of a maganesium fertilization system in wheat production. Unfortunately, such an unequivocal answer cannot be presented directly. The analysis of the obtained data shows that the response of winter wheat to the method of Mg application was very stable in the subsequent years of the study. However, the significant of weather cannot be ignored.
All figures presented in the manuscript were prepared in accordance with the results of the statistical evalution of data. According to ANOVA (Tables 1 and 2), the influence of the Y × Mgs × Mgf interaction on some of wheat charateristics cannot be ignored, as shown in Figures 1, 2, 3, 6.
In order to improve the clarity of the presented data, Figure 1 has been modified. Moreover, the original Figures 1 and 3 have been moved to the appendix. There are no statistical and logical grounds for changing the structure of Figures 2 and 4. The information contained in these two figures is crucial for a reliable assessment of the impact of Mg fertilization systems on N management by winter wheat.
Table 2 should include comparisons of same Mg rates applied to the soil vs applied foliar, for each Mg rates.
The answer to this question in included in the figures showing the interaction Y × Mgs × Mgf for the presented traits of winter wheat. The reader will find data on the value of the given characteristics for the crucial variants: aboslute, Mg control, (only NPK), Mgs25. Mgs50, Mgf-I, Mgf-II, Mgf-I+II.
Looking at the supplemental materials (fertilization schedule table) - not clear how these NPK rates were decided upon -are these typical rates for the area/wheat class? I might have missed this.
As sugegsted by other reviewers, the full reserach methodology was included in the revised version of the manuscript. The forecrop of winter wheat was winter oilseed rape (the best one in the world humid climate areas). The dose of P and K were determined on the basis of soil fertilitym, but its value was established as constant for the entire study period. The N dose was established at the level about 60% of total N uptake by winter wheat at maturity for the yield of 10 t ha-1. In farming practice for this yield is recommended 240 kg N ha-1.
Some rewording might be good throughout - I do recommend an English native speaker or an editor to look the text over. For example: in Conclusion section it is worded as :"Magnesium applied to wheat resulted in a significant yield gain with respect to the effect of NPK.The method of application was of secondary importance." In this case, I would emphasize that the RATE of Mg had a more substantial impact on wheat yield, compared to Mg application method. Also, in statistical methods: "To determine the wheat grain yield, stepwise regression was used to define the optimal set of wheat components."- This statement does not read right, or is part of the statement missing?
A revised version of the manuscript has been checked by the MDPI language editing service. The certificate is attached.

Reviewer 2 Report
Magnesium Fertilization System as a Fertilizing Factor Increasing the Nitrogen Use Efficiency in winter wheat (Triticum aestivum L.)
This paper studied the role of fertilized of Mg in increasing nitrogen fertilizer (Nf) efficiency and yield. This. The work is useful, but there are a lot of problems about the manuscript. The questions from the manuscript is as following.
1) The object of this Ms is not clear, and there is a lack of novelty.
2) The abbreviation should not present in abstract, such as BBCH, and the experiment design in not clear.
3) In experiment design part, experiment setup is not clear, we don’t know how many treatments and when to carry out?
4) The yield is very low, why is so low?
5) The table and figure should be self-evident, but there is no note about I, I+II, IBBCH-30, II—BBCH 49/50.
6) Some formula may be wrong, such as line 433 and 437, and note is not clear.
7) The data may be wrong. The yield is very low as 0.7t (700kg/ha), but the N accumulation is high as 200-250 kg/ha, is it possible?
8) Why N accumulation in 2015 higher than 2014, and higher than 2013 in Fig 4 and 5?
Author Response
Review Report 2 - response
Comments and Suggestions for Authors
Magnesium Fertilization System as a Fertilizing Factor Increasing the Nitrogen Use Efficiency in winter wheat (Triticum aestivum L.)
This paper studied the role of fertilized of Mg in increasing nitrogen fertilizer (Nf) efficiency and yield. This. The work is useful, but there are a lot of problems about the manuscript. The questions from the manuscript is as following.
1) The object of this Ms is not clear, and there is a lack of novelty.
2) The abbreviation should not present in abstract, such as BBCH, and the experiment design in not clear.
3) In experiment design part, experiment setup is not clear, we don’t know how many treatments and when to carry out?
4) The yield is very low, why is so low?
5) The table and figure should be self-evident, but there is no note about I, I+II, IBBCH-30, II—BBCH 49/50.
6) Some formula may be wrong, such as line 433 and 437, and note is not clear.
7) The data may be wrong. The yield is very low as 0.7t (700kg/ha), but the N accumulation is high as 200-250 kg/ha, is it possible?
8) Why N accumulation in 2015 higher than 2014, and higher than 2013 in Fig 4 and 5?
Responding to a review that is extremely subjective, requires drawing the reviewer’s attention to a few mistakes. These errors should not have made assuming the full professional ethics of the reviewer.
- Firstly, the methodology of the research experiment is presented in the suplement. Originally, it was presented in the article as below:
Grzebisz, W.; Potarzycki J. Effect of magnesium fertilization systems on grain yield formation by winter wheat (Triticum aestivum L.) during the grain filling period. Agronomy 2022, 12, 12.
As sugegsted by other reviewers, the full reserach methodology was included in the revised version of the manuscript.
- Secondly, while reviewing the articles on response of crops to Mg fertilization, I did not find any article in which the methodology took into account the interaction of soil and foliar effect of Mg fertilization to winter wheat. The only exception is my research team.
Therefore, on what basis was the conclusion about the lack of scientific novelty of the manuscript submitted to Plants. When making such a request, the reviewer is obligated to present evidences.
When discriminating against the scientific level of the authors, the reviewer should become acquainted with their scientific achievements. I present two of my materials in this area:
- Grzebisz, W. Crop response to magnesium fertilization as affected by nitrogen supply. Plant Soil 2013, 368, 23–39.
- Grzebisz, W. Magnesium. In: Handbook od Plant Nutrition. (Eds.) Barker, A.V., Pilbeam, D.J., CRC Press, 2015, 199-260.
- Thirdly, review points 4 and 7 indicate that the reviewer did not read carefully the content of the text, figures and tables.
The title of the ordinate axis in Figure 1 is: Yield gain!!!!!!!!! That is, an increase in the yield. The title of the figure sounds:
Figure 1. The increase in the yield of winter wheat in response to interaction of Mg fertilization systems and years.
Wheat grain yield ranged from 7.9 to 11.18 t ha-1. In the first variant, the yield gain was 0,52 t ha-1, and in the second 0.98 t ha-1. The obtained the increase in the yield depended on the amount of extra N taken by wheat fertilized with Mg.

Reviewer 3 Report
Title: Magnesium fertilization system as a fertilizing factor increasing the nitrogen use efficiency in winter wheat (Triticum aestivum L.)
The study reported a field experiment that aims to explore the linking between Mg fertilization and N uptake and NUE of wheat based on the application methods and dose. The experiment method and treatment have been showed in other work, but I think the readers would like to see a global presentation of these parameters in the manuscript.
Introduction
1. The first sentence should be shortened.
2. This part should be strengthened the linkage between Mg and N on the effect that promotes crop uptake and utilization of nutrients.
3. Magnesium fertilization system should be explained.
4. L48&56: Please modified the font of “critical window” and “forgotten”.
5. L52-53: This sentence should be modified.
Results
1. I recommend merging these results to keep the core points in every parts.
2. It would better to summary the interaction of Mg dose and nitrogen accumulation and use efficiency.
3. “i.e.” was employed too much, please replace it.
Discussion
1. I suggest that strengthen the discussion of effect of Mg dose and the development stage of wheat on N uptake.
2. It would better to present the main factors affecting the N uptake and NUE of wheat from method and time of application, Mg dose and years.
Materials and Methods
1. I think the readers would like to see a global presentation of these parameters, so these necessary details should be presented in the study: experiment treatments, replicates, soil conditions, and fertilization dose.
2. Please explain the abbreviation meanings, such as “GY”, “GYNi”.
3. I suggest that put the units (e.g., kg ha-1) after the meaning of formula.
4. L440: “[45]” should be deleted.
Author Response
Review report 3 - response
The study reported a field experiment that aims to explore the linking between Mg fertilization and N uptake and NUE of wheat based on the application methods and dose. The experiment method and treatment have been showed in other work, but I think the readers would like to see a global presentation of these parameters in the manuscript.
The full study methodology is included in the Materials and Methods section (Part 4).
Introduction
- The first sentence should be shortened.
This sentence has been shortened.
- This part should be strengthened the linkage between Mg and N on the effect that promotes crop uptake and utilization of nutrients.
The introduction was enriched with a part concerning the effect of the Mg × N interaction on crops.
The interaction between Mg and N occurs at all levels of a plant’s organization. The importance of N for a plant’s growth and yield results from its presence in key biological molecules such as chlorophyll and the ribulose bisphosphate carboxylase-oxygenase enzyme, simply called Rubisco (RuBP) [22]. The latter is a key N-dependent enzyme, decisive in the survival of life on Earth. Its key function is to capture and then fixate the CO2 molecule, which is the basic substrate for the production of elementary sugar compounds [23, 24]. The key function of Mg is to maintain Rubisco activation, which results in the stabilization of the net photosynthetic rate, as was demonstrated for wheat by Shao et al. [25]. Crop plants well supplied with Mg since the beginning of their growth increases N uptake, resulting in the increase of its unit productivity [26, 27]. For example, Mg concentration in maize leaves during the grain filling period is the critical factor affecting the grain yield of this crop. The adequate nutrition of plants with Mg increases N productivity, in turn decreasing the required Nf dose [28, 29].
- Magnesium fertilization system should be explained.
The concept of „Mg fertilization systems” has been described and explained in a separate section of the Introduction”.
- L48&56: Please modified the font of “critical window” and “forgotten”.
Italics has been intentionally used for both terms to emphasize the yield-forming role of Mg. The term critical window as explained in the text refers to the critical stages in the formation of yield components by winter wheat. The term in sucha n approach is laso often used in USA (maize).
The term forgotten is used too often by researches. This is not due to a lack of knowledge about a given nutrient fucntion in the plant, but due to a lack of fertilization recommendations. This is exactly the case with Mg.
- L52-53: This sentence should be modified.
This sentence has been modified and sounds as follows:
„New, high-yielding wheat varieties, in order to exploit their yield potential, require,
first of all, the development of efficient technologies aimed at the the effective use of
nitrogen fertilizer (Nf).”
Results
- I recommend merging these results to keep the core points in every parts.
This chapter was corrected, paying attention to the aspects of the impact of the Mg fertilization systems on N management by winter wheat. The most important part that has been corrected concerns the yield gain due to Mg application to wheat. It sounds as follows:
The yield gain due to the application of Mg to winter wheat was the result of the interaction between the soil and the foliar treatment (Figure 1). Detailed information on the grain yield and the elements of the yield structure can be found in the article by Grzebisz and Potarzycki [32]. In all years of the study, Mg application, regardless of the Mg fertilization system, increased the grain yield. The recorded increase ranged from approximately 0.58 to 0.74 t ha─1. The lower yields were due to the fact of unfavorable weather conditions during the spring growing season (2013 and 2015). Nevertheless, the effect of the Mg fertilization system (Mg-FS) on the yield gain in the consecutive years of the study was highly stable (Figure A1). A strong interaction between the Mg-FSs was observed (Figure 1). The increase in yield due to the soil-applied Mg (Mgs) in comparison with the absolute Mg control (the plot treated only with NPK) amounted approximately to 0.52 t ha─1. The effect of a single stage of Mg foliar treatment (Mgf), regardless of the growth stage of wheat, was 0.57 t ha─1. A double stage of Mg foliar application (at BBCH 30 and repeated at BBCH 49/50) resulted in a yield increase of 0.74 t ha─1. The effect of the interaction of both Mg-FSs on the yield gain was dependent on the dose of Mgs. The increase in yield was 0.77 t ha─1 on the plot fertilized with 25 kg Mg ha-1, but it provided Mg foliar application at BBCH 30. The same level of increase in the yield was recorded on the plot with 50 kg Mg ha-1, regardless of the growth stage of wheat treated with Mg. In-soil and double, two-phase foliar feeding of wheat with Mg resulted in a yield gain of 0.92 t ha─1.
Figure 1. The net increase in the yield of winter wheat in response to the interaction of Mg fertilization systems. Mgs and Mgf—soil and foliar Mg fertilization systems, respectively. * Doses of applied Mg, kg ha‒1; ** stages of Mg. Wheat stages of Mg foliar fertilization: I—BBCH 30; II—BBCH 49/50.
The remaining changes were included directly in the corrected text. Parts of the chapter results, causing infomation noise have been removed from the text.
- It would better to summary the interaction of Mg dose and nitrogen accumulation and use efficiency.
This is what has been done in the Results and Discussion sections. The interaction of Mg and N is prezented in three figures (A1, A2 and A3).
- “i.e.” was employed too much, please replace it.
This shortcut was corrected throughout the manuscript.
Discussion
- I suggest that strengthen the discussion of effect of Mg dose and the development stage of wheat on N uptake.
The discussion has been worked through. The attention was paid to significant aspects of fertilization systems on N management in winter wheat.
- It would better to present the main factors affecting the N uptake and NUE of wheat from method and time of application, Mg dose and years.
The effect of the years, as well as of individual Mg fertilization systems, including their interaction, has been thoroughly presented and discussed. The essence of the issue is presented in this part of the text, included in the discussion:
The course of weather during the growing season did not change much the trend of the wheat response to the tested Mg fertilization systems. On this basis and according to the data in the literature, three strategies for wheat fertilization with Mg during growth can be identified:
- Conservative: a high and stable yield increase, increasing the resistance to abiotic stresses;
- Effective: a moderate to high yield increase, provided there are no abiotic stresses;
- Prophylactic: a moderate yield increase, with a constant fertilization factor.
The first strategy was very efficient in years with some disturbances in the weather course during the spring growing season of winter wheat (Figure 1). It requires, however, a higher dose of the in-soil applied Mg and its frequent application to wheat foliage [14, 19, 20]. The yield-forming effect of the applied Mg can be explained both by the increase in the GD and the maintenance of the photosynthetic activity of leaves during the grain-filling period [24, 25, 32]. The second Mg fertilization strategy can be recommended in the years or regions of the world with favorable growth conditions for winter wheat. A sufficiently high yield increase can be achieved by applying relatively low in-soil Mg doses, provided a high solubility of fertilizer is used as well as one-stage foliar fertilization with Mg [19, 26]. The third fertilization strategy of winter wheat with Mg is based on a double-stage foliar fertilization, regardless of the weather and soil conditions [27].
Materials and Methods
- I think the readers would like to see a global presentation of these parameters, so these necessary details should be presented in the study: experiment treatments, replicates, soil conditions, and fertilization dose.
As sugegsted by the reviewer, the full reserach methodology was included in the revised version of the manuscript. The forecrop of winter wheat was winter oilseed rape (the best). The dose of P and K were determiend on the basis of soil fertility. The N dose was fixed at the level about 60% of thal N uptake by winter wheat at maturity for the yield of 10 t ha-1. In farming practice for this yield is recommed 240 kg N ha-1.
- Please explain the abbreviation meanings, such as “GY”, “GYNi”.
Both terms have been corrected in the text.
- I suggest that put the units (e.g., kg ha-1) after the meaning of formula.
This has been completed.
- L440: “[45]” should be deleted.
It has been deleted.

Reviewer 4 Report
In the Material and methods section, a field experiment scheme should be presented. This will make it easier to read the article quickly without having to refer to supplements.
l. 54 - is: 2021 and 2002; it should be 2021 and 2022
Author Response
Review report 4_response
Comments and Suggestions for Authors
In the Material and methods section, a field experiment scheme should be presented. This will make it easier to read the article quickly without having to refer to supplements.
As also sugegsted by other reviewers, the full reserach methodology was included in the revised version of the manuscript. The forecrop of winter wheat was winter oilseed rape (the best).
- 54 - is: 2021 and 2002; it should be 2021 and 2022
The English language editing certificate is attached.
These errors have been corrected.

Round 2
Reviewer 3 Report
This version can be accepted.